# *Bracon brevicornis* Genome Showcases the Potential of Linked-Read Sequencing in Identifying a Putative *Complementary Sex Determiner* Gene

**DOI:** 10.3390/genes11121390

**Published:** 2020-11-24

**Authors:** Kim B. Ferguson, Bart A. Pannebakker, Alejandra Centurión, Joost van den Heuvel, Ronald Nieuwenhuis, Frank F. M. Becker, Elio Schijlen, Andra Thiel, Bas J. Zwaan, Eveline C. Verhulst

**Affiliations:** 1Laboratory of Genetics, Wageningen University & Research, 6708PB Wageningen, The Netherlands; bart.pannebakker@wur.nl (B.A.P.); joost.vandenheuvel@wur.nl (J.v.d.H.); frank.becker@wur.nl (F.F.M.B.); bas.zwaan@wur.nl (B.J.Z.); 2Population and Evolutionary Ecology Group, Institute of Ecology, FB02, University of Bremen, 28359 Bremen, Germany; alecenturiion@gmail.com (A.C.); thiel@uni-bremen.de (A.T.); 3Bioscience, Wageningen University & Research, 6708PB Wageningen, The Netherlands; ronald.nieuwenhuis@wur.nl (R.N.); elio.schijlen@wur.nl (E.S.); 4Laboratory of Entomology, Wageningen University & Research, 6708PB Wageningen, The Netherlands; eveline.verhulst@wur.nl

**Keywords:** 10X Genomics, Braconinae, complementary sex determination, microsynteny, *feminizer*, parasitoid, parasitic wasp

## Abstract

*Bracon brevicornis* is an ectoparasitoid of a wide range of larval-stage Lepidopterans, including several pests of important crops, such as the corn borer, *Ostrinia nubilalis*. It is also one of the earliest documented cases of complementary sex determination in Hymenoptera. Here, we present the linked-read-based genome of *B. brevicornis*, complete with an ab initio-derived annotation and protein comparisons with fellow braconids, *Fopius arisanus* and *Diachasma alloeum*. We demonstrate the potential of linked-read assemblies in exploring regions of heterozygosity and search for structural and homology-derived evidence of the *complementary sex determiner* gene (*csd*).

## 1. Introduction

*Bracon brevicornis* (Wesmael) is a gregarious ectoparasitoid of various Lepidoptera larvae, including many important pests, and is considered a cosmopolitan species [1,2]. In the past, *B. brevicornis* has been classified under the genus *Habrobracon* [3], *Microbracon* [4], or classified as one species with *Habrobracon/Bracon hebetor* [5]; however, recent research shows that *B. brevicornis* and *B. hebetor* are genetically two distinct species [6]. In the field, *B. brevicornis* has shown potential as a biological control agent against important pest species in stored corn stalks, such as *Ostrinia nubilalis* and *Sesamia cretica* [7], or against the coconut moth, *Opisinia arenosella* [2]. In the laboratory, *B. brevicornis* attacks a wide range of larval host such as *Ephesthia kuehniella*, *Galleria* spp., and *Spodoptera* spp. [8].

Work on *B. brevicornis* has included both laboratory and semi-field set-ups to determine both its efficacy as a biological control agent as well as its suitability as a study system. There are several studies on the biology of *B. brevicornis*, including on population growth potential [9], their host range [8], interspecific competition [2], clutch size and fitness [10], mate choice [11,12], diet [1], and efficacy [7].

Within a phylogenetic perspective, *B. brevicornis* falls within the subfamily Braconinae, the largest of the cyclostome-forming braconid wasps [13]. The presence of a cyclostome (round mouthpart) is a defining feature within braconid wasps, as it represents an unresolved evolutionary and systematic question: is the cyclostome a derived trait within certain branches, or an ancestral trait that has been lost in others [13]? Within the Braconinae, there have been multiple switches from ectoparasitism to endoparasitism and vice versa, and this combination of cyclostome and endoparasitism has been described as a “controversial topic” by braconid researchers and taxonomists [13]. These systematic issues are far from being resolved, and more genomic data would be useful for future phylogenetic analyses [13]. Yet, a representative genome for the Braconinae is currently lacking. As previously stated, *B. brevicornis* is an ectoparasitoid, and its position within a family that contains both types of parasitism lifestyles holds promise for further phylogenetic comparisons.

As a member of the order Hymenoptera, *B. brevicornis* has a haplodiploid sex determination system where males develop from unfertilized eggs and females develop from fertilized eggs [14,15,16]. From a genetic perspective, *B. brevicornis* belongs to a genus where sex determination and diploid male production have been widely studied (*B. hebetor* [17], *B. brevicornis* [3], *B. serinopae* [18], reviewed in [19]; *B.* spec. near *hebetor* [20]; *B. variator* [21]). Indeed, the first description of the complementary sex determination (CSD) mechanism was provided for *B. hebetor* (= *B. juglandis* [22], reviewed in [23]), and recent work on *B. brevicornis* and polyploidy studies include diploid male fitness as well as ploidy-dependent mate choice behavior [11].

While straightforward to detect phenotypically through the formation of diploid males following inbreeding [14,19], the molecular mechanism underlying CSD has thus far only been resolved to a small level of detail in the honeybee *Apis mellifera* (L.) (Hymenoptera: Apidae), with the identification of the *complementary sex determiner* (*csd*) gene [24,25]. Heterozygosity at this gene leads to female development, while hemi- and homozygous individuals develop into haploid and diploid males, respectively. Therefore, inbreeding often leads to diploid male production in species with a CSD mechanism as it increases homozygosity. *Csd* is a duplication of *feminizer* (*fem*), a *transformer* (*tra*) ortholog [26] that is conserved across many insect orders as part of the sex determination cascade [27]. When heterozygous, *csd* initiates the female-specific splicing of *fem*, which then autoregulates its own female-specific splicing, ultimately resulting in female development [26,28]. Within the Hymenoptera and specifically the bees and ants, more duplications of *tra*/*fem* have been identified in species that are presumed to have CSD [27], but these *tra*/*fem* duplications have not been analyzed for potential heterozygosity. There has been some debate on the origin of *tra/fem* duplications. One theory states that the duplication of *fem* to produce *csd* occurred before the divergence of ants and bees, and that the paralogs evolved in concerted evolution [29,30], while the other theory states that all *fem* duplications are independent events [31,32]. This shows that additional hymenopteran genomes are necessary to understand the evolutionary history of *tra*/*fem* duplications and identify the genes underlying CSD. However, an assembled genome is usually haploid as areas of heterozygosity are collapsed in the final stages of assembly, diminishing the potential to find the gene(s) underlying CSD by its most notable feature: heterozygosity. Yet recent advances in sequencing and analysis gave us the ability to view heterozygous regions—known as “phases” in diploid assemblies—within a genome, which allow us to investigate the genes potentially involved in CSD.

Here, we report on the whole-genome sequencing of a pool of females from an isolated *B. brevicornis* strain using 10X Genomics technology that relies on linked-read sequencing (10X Genomics Inc., Pleasanton, CA, USA). Due to their long history of genetic isolation during laboratory rearing, individuals in this strain are assumed to have a high level of homozygosity, whereas a *csd* locus would retain its heterozygosity in the females. The 10X Genomics technology allows for generating phased data in which allelic variants can be identified after assembly. High-molecular weight DNA is partitioned into small droplets containing a unique barcode and adapter in such a way that only a few DNA molecules are present within each droplet.

Within each droplet (Gel Bead-in-Emulsion, “GEM”) the DNA is used as template for random priming and elongation resulting in many short DNA fragments sharing the same unique barcode. These DNA fragments are converted into a library that can be sequenced on an Illumina sequence platform. In the assembly step, the reads originating from the same fragment are organized by barcode and put together into synthetic linked-read fragments. Importantly, it is nearly impossible that two fragments with opposing allelic variants are together in the same droplet [33], though smaller genomes are a challenge on their own. This technique therefore allowed us to identify potential *csd* candidates in the female-derived *B. brevicornis* genome after sequencing by studying the phased data containing the different haplotypes. Moreover, as *B. brevicornis* is a potential biological control agent of several pests, the availability of a full genome may provide effective ways to study and improve this species to grow it into an established biological control agent for Lepidopteran pests.

## 2. Materials and Methods

### 2.1. Species Description and General Rearing

Individuals of *B. brevicornis* were taken from the laboratory colony L06. The colony was initiated in 2006 from naturally parasitized *O. nubilalis* larvae collected in maize fields near Leipzig, Germany. Species identification was first carried out by Matthias Schöller and Cornelis van Achterberg based on morphological characteristics [34]. Since collection, parasitoids have been reared on late instar larvae of the Mediterranean flour moth, *E. kuehniella* [11]. The species identity of strain L06 was recently revalidated based on molecular data and is entirely separate from its congeneric *B. hebetor* [6].

### 2.2. DNA Extraction

Immediately following emergence, 100 to 120 female wasps were flash frozen in liquid nitrogen and ground with a mortar and pestle. Genomic DNA was extracted using a protocol modified from Chang, Puryear, and Cairney [35]. Modifications include adding 300 μL β-mercaptoethanol (BME) to extraction buffer just before use. Instead of 10M LiCl, 0.7 volume isopropanol (100%) was added to the initial supernatant, after which it was divided into 1.5 mL Eppendorf tubes as 1 mL aliquots for subsequent extractions. The initial centrifugation step occurred at a slower rate and for a longer period of time to adjust for machine availability. Final pellets were dissolved in 50 μL autoclaved MQ H_2_O and recombined at the end of the extraction process (1.0 mL). DNA concentration was measured with an Invitrogen Qubit 2.0 fluorometer using the dsDNA HS Assay Kit (Thermo Fisher Scientific, Waltham, MA, USA) with final assessments for DNA quality, amount, and fragment size confirmed via BioAnalyzer 2100 (Agilent, Santa Clara, CA, USA).

### 2.3. 10XGenomics Library Preparation and Sequencing

As the genome of *B. brevicornis* is relatively small for the scale of the 10X platform, there is a higher risk of overlapping fragments within single GEMs. In order to reduce this risk, genomic DNA of a larger and previously analyzed genome (Tomato, *Solanum lycopersicon* (L.) (Solanaceae), commercial variety Heinz 1607) [29] was used as “carrier DNA”. DNA extraction of *S. lycopersicon* followed the protocol of Hosmani et al. [36]. The DNA of both *B. brevicornis* and *S. lycopersicon* was pooled in a 1:4 molar ratio.

One nanogram of this pooled DNA was used for 10X Genomics linked-read library preparation following the Chromium Genome Reagent Kits Version 1 User Guide (CG-00022) (10XGenomics, Pleasanton, CA, USA). Barcoded linked-read DNA fragments were recovered for final Illumina library construction (Illumina, San Diego, CA, USA). The library was used for 2 × 150 bp paired-end sequencing on one lane of an Illumina HiSeq 2500 at the business unit Bioscience of Wageningen University and Research (Wageningen, The Netherlands). Sequencing data were then used for basecalling and subsequent demultiplexing using Longranger (v2.2.2) (10X Genomics) (command—mkfastq), yielding 212,910,509 paired-end reads with a read length of 150 bp.

### 2.4. Assembly

To filter sequence data from tomato carrier DNA, 23 bp (16 bp GEM + 7 bp spacer) was removed from forward reads and all reads were subsequently mapped to an in-house high quality reference assembly of the Heinz genome using BWA-MEM v0.7.17 [37]. Using samtools v1.9 [38], all unaligned read pairs (-F = 12) were extracted and labelled non-Heinz. The assembly of the non-Heinz labelled read set was performed with Supernova assembler v2.1.0 (10X Genomics), using default settings including commands for both pseudohap (--style = pseudohap) and pseudohap2 (--style = pseudohap2) outputs [33]. These commands determine the output from Supernova, the first being the final scaffold output (pseudohap), while the second is the so-called “parallel pseudohaplotype” (pseudohap2) scaffolds that represent areas of divergence or phases [33]. Phasing is flattened in the pseudohap output by selecting the region with higher mapping coverage, whereas in the pseudohap2 output is differentiated by “.1” and “.2” at the end of each scaffold name to denote phasing, though not all scaffolds are phased at this point due to lack of divergence during assembly.

To verify whether there were no Heinz leftovers in the assembly, minimap2 v2.17-r941 [39] was used to align the assembly against the Heinz reference assembly. Further examination on presence of possible non-*B. brevicornis* scaffolds, i.e., bacterial scaffolds from sample microbiome, was performed with BlobTools (v1.0) [40], relying on megaBLAST against the NCBI NT-NR database [41] (2018-11-19) (max_target_seqs = 1, max_hsps = 1, evalue = 1 × 10^−25^) for taxonomical classification and BWA-MEM mapping of reads against scaffolds for coverage statistics. Reads mapping only against “Arthropoda” classified scaffolds were then extracted and used for a final k-mer analysis using jellyfish v2.1.1 (-C m = 21 –s = 2,000,000,000) [42] and GenomeScope [43] to infer heterozygosity.

Assembly completeness was determined using BUSCO (v3.0.2) with the insect_odb9 ortholog set and the fly training parameter [44] while assembly statistics were determined using QUAST [45]. The aforementioned pseudohap2 scaffolds were used in *csd* analysis, while the pseudohap scaffolds are now the assembly used for annotation.

### 2.5. Ab Initio Gene Finding and Protein Comparison

The coding sequences of two additional braconids (members of the subfamily Opiinae, and similar to the Braconinae belonging to the cyclostome subgroup [13,46]) were used for gene prediction and protein comparisons: *Fopius arisanus* (Sonan) (Hymenoptera: Braconidae) and *Diachasma alloeum* (Muesebeck) (Hymenoptera: Braconidae). Both sets of coding sequences were retrieved from the NCBI Assembly Database, version ASM8063v1 for *F. arisanus* and version Dall2.0 for *D. alloeum* [41,47,48].

For gene prediction, Augustus (v2.5.5) was first used to predict genes from the *B. brevicornis* assembly [49]. Using BLAST, coding sequences of *F. arisanus* were set as a query to the genome of *B. brevicornis* using default parameters (except minIdentity = 50) [50]. The result was converted into a hints file that was used to predict the genes of *B. brevicornis* using *Nasonia vitripennis* (Walker) (Hymenoptera: Pteromalidae) as the species parameter in Augustus (--species = nasonia –extrinsiccCfgFile = extrinsic.E.cfg).

After prediction, the protein sequences were retrieved and compared to both *F. arisanus* and *D. alloeum* (version Dall2.0) using Proteinortho (v6.0, -p = blastp, -e = 0.001) [51]. From the orthology grouping generated by Proteinortho, gene names could be allocated to the predicted genes. Lengths of both of these *B. brevicornis* genes and the orthologs of *F. arisanus* and *D. alloeum* were retrieved using samtools for comparison [38]. The mean relative length of genes was retrieved by dividing the lengths of all aorthologous amino acid sequences between the two relevant species and then calculating the mean over these. Errors within the annotation related to genome submission and validation were corrected with manual annotation of exons (three cases) and removal of two predicted genes that were more than 50% ambiguous nucleotides.

### 2.6. In Silico Identification of Feminizer as a Putative csd Locus

The pseudohap2 files were deduplicated using the dedupe tool within BBTools (sourceforge.net/projects/bbmap/) (ac = f) to remove all parallel pseudohaplotypes that were complete duplicates as these scaffolds were not heterozygous. The remainder of the set contained both scaffolds that previously had a duplicate, as well as solitary scaffolds that did not have a partner scaffold. These unique scaffolds were removed using the “filter by name” tool in BBTools, leaving 258 scaffolds, or 129 pairs of pseudohap2 scaffolds. Pairs were pairwise aligned in CLC Genomics Workbench 12 (Qiagen, Hilden, Germany) using default settings (gap open cost = 10, gap extension cost = 1, end gap cost = free, alignment = very accurate).

A local tBLASTn search against the entire *B. brevicornis* assembly was performed using the *Apis mellifera* Feminizer protein (NP_001128300) as query in Geneious Prime 2019.1.3 (http://www.geneious.com, [52]). The protein of gene “g7607” (locus tag = BBRV_LOCUS33129) was used in an NCBI BLASTp against the nr database with default settings [41,50]. Next, a region stretching from ~10 Kbp upstream and downstream of the first and last tBLASTn hit in scaffold 12, respectively, was annotated using a hidden Markov model (HMM) gene finder plus similar protein-based gene prediction (FGENESH+, Softberry, http://www.softberry.com/) with *Nasonia vitripennis tra* (NP_001128299) and *N. vitripennis* for the specific gene-finding parameters [53]. Only this combination of settings resulted in a full-length annotation from TSS to poly-A with seven exons. The resulting protein prediction was used in a BLASTp search with default settings against the nr database. To annotate the potential *fem* duplication, a stretch of ~10 Kbp directly upstream of the annotated putative fem was again annotated using FGENESH+ (Softberry) with *Nasonia vitripennis tra* (NP_001128299) and *N. vitripennis* for the specific gene-finding parameters [53]. The predicted annotation contained five exons but lacked the last coding segment with stop codon. A protein alignment was made in Geneious Prime 2019.1.3 with *A. mellifera csd* (ABU68670) and *fem* (NP_001128300)*; N. vitripennis tra* (XP_001604794) and *B. brevicornis* putative *fem* and *B. brevicornis* putative *fem* duplicate (*fem1*), using MAFFT v7.450 with the following settings: Algorithm = auto, Scoring matrix = BLOSUM62, Gap open penalty = 1.53, Offset value = 0.123 [54,55].

### 2.7. Microsynteny Analysis

A microsynteny analysis was achieved by comparing the arrangement of a set of homologous genes directly upstream and downstream of *tra* or *fem* in *A. mellifera* and *N. vitripennis* using a combination of the online tool SimpleSynteny [56] and tBLASTn searches using default settings in Geneious Prime. The scaffolds containing *fem* (*A. mellifera,* scaffold CM000059.5, 13.2 Mbp in length), *tra* (*N. vitripennis,* scaffold NW_001820638.3, 3.7 Mbp in length) or the putative *fem* (*B. brevicornis,* scaffold 12, 4.5 Mbp in length) were extracted from their respective genomes (*Apis*: GCA_000002195.1_Amel_4.5_genomic, *Nasonia*: nvi_ref_Nvit_2.1, *Bracon*: *B. brevicornis* assembly from this study) and searched with protein sequence from the following genes: *tra* (GeneID: 00121203), LOC100121225, LOC100678616, LOC100680007 originating from *N. vitripennis*; *fem* (GeneID:724970), *csd* (GeneID:406074), LOC408733, LOC551408, LOC724886 originating from *A. mellifera*. The advanced settings for SimpleSynteny were as follows: BLAST E-value Threshold = 0.01, BLAST Alignment type = Gapped, Minimum Query Coverage Cutoff = 1%, Circular Genome Mode = Off. If the gene was not found within the extracted scaffold, it was searched for in the full genome assembly. For the image settings, Gene Display Mode = Project Full-Length Gene. This generated image was used together with results from the tBLASTn searches as template to draw the final figure. The final figure that we present in the Results and Discussion section depicts ~0.9 Mbp of genomic region for all three species.

### 2.8. Data Availability

Raw sequence data for *B. brevicornis* after removal of carrier DNA and contamination, as well as the assembly, can be found in the EMBL-EBI European Nucleotide Archive (ENA) under BioProject PRJEB35412. Both the assembly file (.fasta) (https://doi.org/10.6084/m9.figshare.12674189.v2) and the complete annotation file (.gff) (https://doi.org/10.6084/m9.figshare.12073911.v2) are available in a separate repository. Contaminated pseudohap scaffolds are available for download alongside the two pseudohap2 FASTA files, more details are provided at https://doi.org/10.17026/dans-xn6-pjm8.

## 3. Results

### 3.1. Genome Assembly

A total of 172 ng of *B. brevicornis* DNA was extracted, which was then reduced to 1 ng/μL for library preparation. Sequencing of the Heinz diluted library resulted in a total yield of 54 Gbp of data (corrected for 10X Genomics 23 bp segment of forward reads). Mapping against the Heinz genome assembly showed a mapping percentage of 84.8%. There was a total of 30,278,915 unmapped pairs, comprising ~8.39 Gbp of data. This corresponds to the 4:1 ratio between Heinz and *B. brevicornis* DNA in the library. Further scaffold decontamination with BlobTools resulted in a separation of the assembly into *B. brevicornis* scaffolds and microbiome scaffolds. The final assembled genome is ~123 Mbp in size, comprised of 353 scaffolds. This is similar to the projected physical genome size of 133 Mbp ([57], flow cytometry). The contig N50 of the assembly is 6,121,327 bp, while k-mer analysis of the *B. brevicornis-*only read set showed an expected haploid genome length of ~115 Mbp (105 Mbp unique, 10 Mbp repeat) and a heterozygosity of ~0.54%. Peak coverage was 27x. The disparity in projected physical genome size and assembled genome size may be due to the decontamination process potentially removing extremely repetitive regions, as ~10.9 Mbp of the removed scaffolds returned either no hits or were classified as undefined at the taxa level. These scaffolds were clearly separated in the blobplot due to their high coverage. The potential effects are negligible, as the BUSCO analysis indicates a completeness of 98.7% (single orthologs 97.0%, duplicate orthologs 1.7%).

### 3.2. Ab Initio Gene Finding and Protein Comparison

In total, 12,686 genes were predicted, with an average coding sequence length of 529.86 amino acids. The number of genes correspond well to those found in *F. arisanus* (11,775) and *D. alloeum* (13,273), the two closest relatives of *B. brevicornis* for which public data are available. Proteinortho analysis resulted in 7660 three-way orthology groups (7830 *B. brevicornis* genes), while 362 orthology groups contained proteins of *B. brevicornis* and *F. arisanus* (382 *B. brevicornis* genes), and 451 groups contained *B. brevicornis* and *D. alloeum* genes (479 *B. brevicornis* genes). A large number of orthology groups (2492) had no *B. brevicornis* genes, while 3995 predicted genes remain ungrouped.

Compared to *F. arisanus*, the mean relative length of predicted *B. brevicornis* genes was 1.016, while the mean relative length for the two- and three-way orthology groups was 0.996. Similar results were obtained for comparisons to *D. alloeum,* where mean relative length for *B. brevicornis* genes was 1.011, and 0.988 for the two- and three-way orthology groups. Furthermore, the pairwise lengths of all these proteins are fairly similar (Figure 1).

### 3.3. Identification of a Putative Feminizer Ortholog and Duplication Event

After deduplicating the similar parallel pseudohaplotype files, 6706 scaffolds in total, the remainder of the set contained 3420 scaffolds, of which 3286 scaffolds were solitary and did not have a counterpart pseudohap2 for comparison. Some had a previous duplicate removed in the deduplication, while others never had a partner scaffold in the first place. These unique scaffolds were removed, leaving 258 scaffolds, or 129 pairs of pseudohap2 scaffolds. These putatively heterozygous scaffolds were good candidates to search for potential *csd* loci as these are presumed to be heterozygous in females.

So far, a *csd* gene has been sequenced only in species of bees of the genus *Apis,* and it is highly polymorphic, even within subspecies [58]. It is located adjacent to the more conserved *feminizer* (*fem*) [26], and we therefore started with localizing *feminizer* in the genome. As *feminizer* (or its ortholog *transformer*, *tra*) was not identified in the ab-initio annotation, we used a local tBLASTn search to find *fem* in the assembly. Four hits with E-value from 5.86 × 10^−4^ to 8.59 × 10^−8^ were found in scaffold 12. Searching the annotation using part of the tBLASTn result shows that it is annotated as “g7607” (locus tag = BBRV_LOCUS33129) which gave a first hit with protein O-glucosyltransferase 2 (*Diachasma alloeum*) after a BLASTp search, and no *fem* or *tra* hits were found. A closer inspection showed that “g7607” is annotated as fusion protein with the N-terminal part resembling *fem* and the C-terminal part putatively encoding *O-glucosyltransferase 2*. Next, we used FGENESH+ to re-annotate the genomic region, resulting in a full-length putative *B. brevicornis feminizer* (*Bbfem*) ortholog containing seven exons (Figure 2). We found that the two *fem*/*tra* signature domains in Hymenoptera, the Hymenoptera domain [59] and CAM domain (putative autoregulatory domain) [60], are present in the putative *fem* ortholog, but are also duplicated upstream of putative *Bbfem*. A second manual re-annotation step showed that a partial *fem*-duplicate is encoded directly upstream of putative *Bbfem* containing five exons (Figure 2), which we denote here as *Bbfem1* as suggested by Koch et al. [31]. The entire region encoding both *Bbfem* and *Bbfem1* has 64.5% pairwise identity (Figure 2). The coding region of *Bbfem* has 83.2%, and *Bbfem1* has 79.3% pairwise identity.

A protein alignment showed that the full-length putative *Bbfem*, as well as *Bbfem1*, are highly similar to each other and both contain all known *fem*/*tra* domains (Figure 3).

*Bbfem1* lacks a notably long first Arginine/Serine (RS)-rich region which is present only in *Bbfem* (124–153aa)*,* but it otherwise appears to encode for a full-length protein. The *csd-*specific hypervariable domain (Figure 3, purple text; [24]) is not present in *Bbfem* nor in *Bbfem1*. Therefore, the gene name has been updated as “g7607 putative Bbfem-Bbfem1 *csd*” in the official annotation.

### 3.4. Synteny Analysis of Putative Fem Encoding Region

We compared the orthologous gene arrangement of a number of genes up- and downstream of *N. vitripennis tra *and *A. mellifera fem* and *csd,* with the genomic organization of the *Bbfem* region (Figure 4).

*N. vitripennis* LOC100680007 is present in the *tra*/*fem* containing scaffolds of all three genomes, while *A. mellifera* LOC408733 has both translocated closer to *Nasonia tra* and to a different scaffold in *B. brevicornis. N. vitripennis* LOC100121225 and LOC100678616 are encoded in opposing directions in both *A. mellifera* and *N. vitripennis* but are both downstream of *tra* in *N. vitripennis* and upstream of *fem* and *csd* in *A. mellifera.* There is no match for both genes in *B. brevicornis*. *A. mellifera* LOC724886 and LOC551408 are encoded in opposing directions with the same orientation in both *N. vitripennis* and *A. mellifera* but are reversed in *B. brevicornis* and downstream of *Bbfem* and *Bbfem1* while they are upstream of *csd* and *fem* in *A. mellifera*. In *N. vitripennis*, both genes are not located in the *tra* containing scaffold but in another scaffold indicating that this region has undergone chromosomal rearrangements.

## 4. Discussion

Here, we present the genome of the braconid wasp *Bracon brevicornis*, a parasitoid wasp that not only has biological control applications, but also offers potential as a study system for future analyses into braconid phylogenetics and gene evolution. With no previous genomes available for the subfamily Braconinae, the most specious of the braconid wasps, the resources and investigations presented here fill this gap. Our linked-read library, assisted by carrier DNA of *S. lycopersicon,* has resulted in a highly contiguous, very complete assembly, comprised of just 353 scaffolds and 12,686 genes. This gene count is similar to related species, and in further protein length comparisons, the proteins are highly similar. This indicates that the predicted genes are highly complete, a necessary feature for any future phylogenetic comparisons between species or families.

We utilized the 10X Genomics linked-read approach to obtain pseudohaploid information that would allow us to search for potential *csd* loci in silico. As a substantial number of scaffolds were putatively heterozygous, we used the notion that in *A. mellifera, csd* is located adjacent to *fem* [26] to limit our search for *csd* candidates. We manually annotated a putative *B. brevicornis fem* and a partial *Bbfem* duplicate that is highly similar, and both genes encode all known *tra*/*fem* protein domains (Figure S1) [59]. Both genes are in a small region that is highly heterozygous, especially when compared to the remainder of the scaffold, which would suggest true heterozygosity and not assembly error, but also when compared to the level of heterozygosity in the other pseudohap2 scaffolds. While the 10X Genomics platform has been discontinued for de novo genome assembly, recent approaches may offer a similar solution (see [61]).

Our synteny analysis showed low synteny between *B. brevicornis*, *A. mellifera*, and *N. vitripennis* with the translocation of LOC408733 (*A. mellifera*) and the absence of LOC100121225 and LOC100678616 (*N. vitripennis*) in the *B. brevicornis* genome region. It is known that genomic regions encoding sex determination genes are dynamic in nature, showing both duplications and translocations [62]. *Tra*/*fem* duplications have been shown in CSD systems before, most notably in *A. mellifera* where a *fem* gene duplication event resulted in it becoming a *csd* locus [26,28], but also in ants and bumblebees, where many independent duplications of *fem* with varying orientations are observed [30]. In addition, in non-CSD systems, *tra* duplications have been observed [27,63,64]. Although there is some debate on whether *fem* paralogs originated due to a single duplication event and functions as *csd* [29,30], or evolved multiple times independently and may have other functions [31,32], we suggest that the *Bbfem* paralog, *Bbfem1*, is a promising *csd* gene candidate in *B. brevicornis*. However, in-depth analyses are required to verify this, including in-depth sequencing of individual females, gene expression analyses, and further phylogenetic studies with other CSD Hymenopterans, all of which are now possible with an assembled, annotated, and published genome.

## Figures and Tables

**Figure 1 genes-11-01390-f001:**
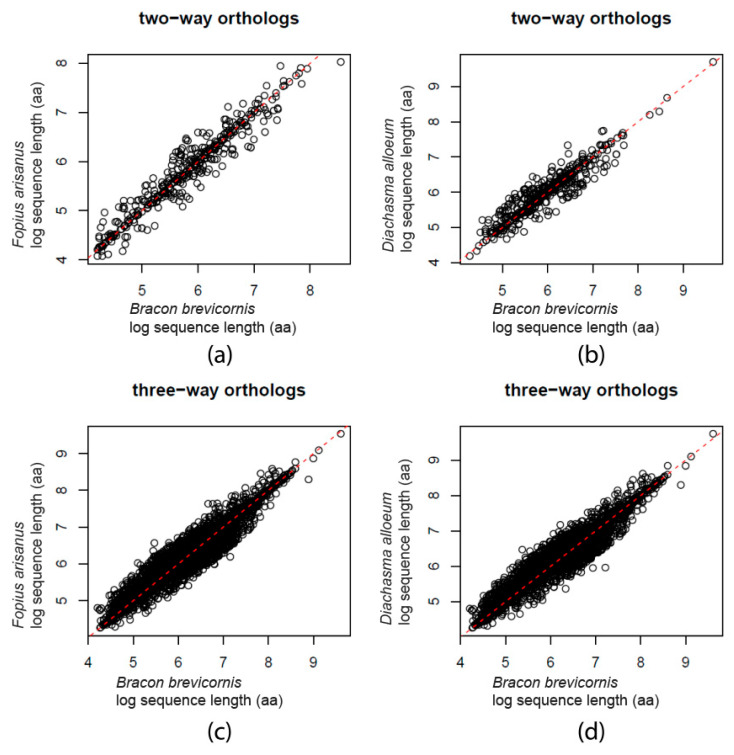
Protein length comparison between *Bracon brevicornis* and *Fopius arisanus*, (**a**) two- and (**b**) three-way orthologs, and *B. brevicornis* and *Diachasma alloeum*, (**c**) two- and (**d**) three-way orthologs. Sequence lengths (amino acids, “aa”) have been log-transformed; red dashed line represents an identity line.

**Figure 2 genes-11-01390-f002:**
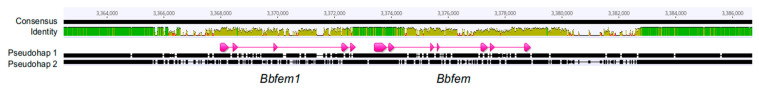
*Bracon brevicornis* annotation of *Bbfem* and *Bbfem1* on the alignment of pseudohaplotype track 1 and 2 in Geneious Prime 2019.1.3 (http://www.geneious.com, [52]). Within the assembled genome, this section corresponds to a region on scaffold 12. The *Bbfem1* annotation lacks the last coding segment with stop codon. The identity track shows the amount of sequence identity across an arbitrary window (depending on zoom setting) and can be used as a proxy for heterozygosity. Green is identical, yellow is mismatch, and red is no match due to introduced gaps during alignment. The coding regions of *Bbfem1* and *Bbfem* are in a high putatively heterozygous region.

**Figure 3 genes-11-01390-f003:**
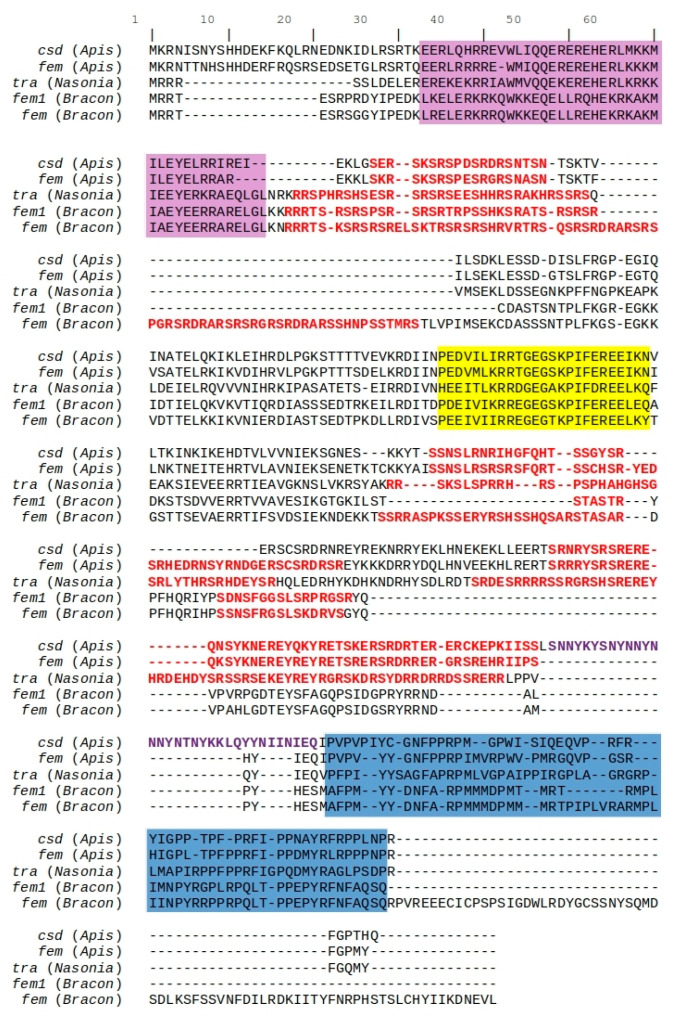
Protein alignment of *Apis mellifera csd* (ABU68670) and *fem* (NP_001128300), *Nasonia vitripennis* tra (XP_001604794), *Bracon brevicornis fem* and *fem1*. Purple shading indicates Hymenoptera domain [59], yellow shading indicates CAM domain [60], blue shading indicates Proline (P)-rich region, red text color indicates Arginine/Serine (RS)-rich regions, and purple text color indicates hypervariable region in *csd* [24].

**Figure 4 genes-11-01390-f004:**
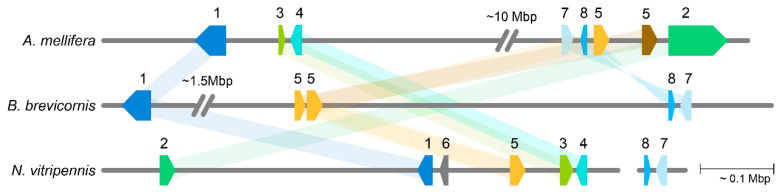
Microsynteny of genomic regions containing *tra/fem* paralogues. Shown is ~0.9 Mbp of the genomic region of *Apis mellifera*, *Bracon brevicornis*, and *Nasonia vitripennis*, containing the approximate coding region for 1. LOC100680007 (dark blue), 2. LOC408733 (green), 3. LOC100121225 (lime), 4. LOC100678616 (cyan), 5. *tra*/*fem*/*fem1* (yellow) and *csd* (brown), 6. LOC107980471 (gray), 7. LOC724886 (blue), 8. LOC551408 (light blue). Locus 2 is located on a different scaffold in *B. brevicornis*, locus 3 and 4 are not present in *B. brevicornis*. Locus 6 is unique to *N. vitripennis*, and locus 7 and 8 are located on a different scaffold in *N. vitripennis*, which is depicted on the right. Both 7 and 8 are in the same order and orientation as in *B. brevicornis* but are reversed in *A. mellifera*.

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
