# Peer review of "Bracon brevicornis Genome Showcases the Potential of Linked-Read Sequencing in Identifying a Putative Complementary Sex Determiner Gene"

_genes, 2020, doi:10.3390/genes11121390_

Round 1

Reviewer 1 Report

Dear authors, 

thank you for updating your manuscript and clarifying some points.  I consider that the actual version of the manuscript is ready for acceptance. 

Just one minor misspelling, in line 189: "all aorthologous amino acid", replace aorthologous by orthologous. 

Hope seeing the continuation of the work in a close future.  

Reviewer 2 Report

The authors have addressed all my concerns. 

Reviewer 3 Report

The authors have responded appropriately to reviewers comments on the first submission of this manuscript. The manuscript will be a notable contribution to the growing genome resources for Hymenoptera. The manuscript also provides preliminary evidence for the identification of a gene involved in sex determination in B. brevicornis while at the same time acknowledges the need to pursue additional analyses to demonstrate these findings more fully. 

This manuscript is a resubmission of an earlier submission. The following is a list of the peer review reports and author responses from that submission.

Round 1

Reviewer 1 Report

Ferguson and coworkers present here a brief insight into the genome of the braconid Bracon brevicornis. The authors describe in deep detail the generation of Illumina libraries, assemblies and mapping to known hymenopteran genomes to obtain a 27x genome. Later on, the authors focus on the genomic region containing "complementary sex determiner, csd" gene to get insights into the sex determination in this insect species.  

Sex determination in insects is an important research field, from its basic nature to the putative applications of the knowledge (improvement of biological control programs outcomes by enhancing the rearing of only females), as stated by the authors in the importance of the subject of study. 

The authors have found that the csd region do not contain any homologous to csd, but contains what could be considered a small scale duplicate of fem (authors name it as Bbfem1). This which seems the main point of the manuscript is also the part that deserves further improvement. The authors have placed on supplementary information some of the phylogenetic trees that I suggest to be moved to the main text, as these corroborates the duplication event of Bbfem, and will add a surplus in the achievement of your future conclusions (lines 363-366). 

Missing literature important to this paper: 

Biewer et al. 2015. Front. Genet. 6:124. doi: 10.3389/fgene.2015.00124(this one is specially interesting, as deals with duplicates of fem).

Cook. 1993. Sex determination in the hymenoptera: a review of models and evidence.  Heredity 71. (or just include along with ref. 14 or when citing 11, for diploid males in B. brevicornis)

Hasseelmann et al 2008. doe:10.1038/nature07052

Zareba et al. 2017.Scientific Reports | 7: 2317 | DOI:10.1038/s41598-017-02629-9

Other points: 

Please clarify whether you are talking about CSD mechanisms and csd gene, in lines 71 to 81.

In the same lines, you stated that csd is a duplication of fem, which is a tra ortholog. It should be a duplicated gene, isn´t it?  You used the reference 24 (Hasselmann et al. Nature 2008) on which they determined that csd was a recent duplicate of fem that occurred in Apis genera. Please check Biewer et al. 2015 for some updates in fem duplicates in Apis, and how these new sequences could be used to determine the duplication event. 

You stated and name the new Bbfem1 as putative csd, could you verify this state by performing a phylogenetic study of the DNA sequence including those fem1 duplicates in Apis? 

Line 168: "D. alloem" should be D. alloeum 

Line 258, "predicted B. brevicornis genes was 1.016"; please place units, maybe kb? or you are talking about aa's?

Lines 265 to 267. figure 1 legend. Place species names into italics. Include the units of sequence length. 

Lines 314-322. Figure 3. Please place the full name of species in the figure instead the genus.  

Lines 323-326. Could you explain where loci 3 and 4 where translocated in B. brevicornis? 

Line 348, a Figure S1 is mentioned but it not appears in the uploaded files for revision. Please check this.  

Reviewer 2 Report

The authors present the genome of the parasitic wasps Bracon brevicornis. They used the recently developed technology 10X Genomics allowing a linked-read assembly, and therefore, the analysis of heterozygous phases of the genome.

By using inbred females, they identified a highly heterozygous region of the genome corresponding to a potential ortholog of the feminizer gene responsible of the sex determination in Hymenoptera. They also identified a duplication of this gene, providing potential insights onto the origin of csd gene in this group.

Overall, the study has been carried out nicely; the results are sound and convincing and the manuscript is interesting.

I only have few minor comments to provide

In the method, I did not clearly understand whether the 100-120 females were pooled during DNA extraction, or whether they were pooled for the Illumina run after being individually barcoded? In the first case, I do not understand how the authors differentiated between heterozygous allelic-variants of a given individual and homozygous allelic-variants between individuals?

L84-86: Rewrite to make clear that all individuals in this strain have a high level of homozygosity, but only females retain heterozygosity at the csd locus.

L94. Change allelic-variances to allelic-variants

L168, 178. Stay consistent between D. alloem or D. alloeum

L206. and should not be italicized

L249. Mbp

L265. Species names should be italicized in the figure caption.

Figure 3. It is not clear why the locus 7 and 8 of N. vitripennis are represented despite being located on a different scaffold, but not the locus 2 of B. brevicornis.

Discussion. Is there any information available on the number of chromosomes in B. brevicornis?

L353. A. mellifera

Reviewer 3 Report

The authors report the sequencing and assembly of the Bracon brevicornis genome. In addition to its potential in biological control of crop pests, the genome of B. brevicornis provides the opportunity to study the evolution of sex determination systems in Hymenoptera. While I believe this newly assembled genome will be a beneficial resource to the scientific community given hymenopteran genomes are underrepresented, I feel that there are limited conclusions that can be made about CSD based on the described analyses. In regards to the search for CSD genes, there is a need for improved clarity in the methods used and also a critical need to confirm the haplotypes assembled are not the result of sequencing/assembly artefacts. Additionally, while the overall manuscript was easy to read/follow, there were some examples of awkward word choice and inconsistent spelling/measurement units that should be rectified before this work is published.    

Below I have included some specific comments on the manuscript, with line numbers corresponding to questions/suggested changes.

Title: How is Braconinae being “revisited” in this manuscript? Not sure if it really adds to the title

Line 42. change “e.g. on” to “including”

Lines 44-55. phylogenetic discussion seems out of context. Especially since it isn’t investigated fully in this manuscript. I think it’s OK to mention the unresolved phylogeny and potential contribution of the B. brevicornis genome to phylogenomic studies in 1-2 sentences in the intro/discussion, but I think this paragraph is distracting.

Line 58. rephrase “interesting genus”, replacing with less subjective terminology

Line 65. shouldn’t “phenotypically” be changed to “genetically”? I’m assuming there aren’t phenotypic differences between haploid and diploid males.

Lines 78-81. Two sentences beginning with “However” could be removed…I don’t know how much they add here and the benefit of linked read sequencing is explained in the following paragraph.

Lines 90-93. I’m not sure if a description of the 10X method is necessary. But one thing you might consider is citing other genome studies that successfully used 10X to phase heterozygous genome regions and thus why it is a particularly attractive solution to your study system.

Lines 93-94. I am not sure what this sentence means, particularly the phrase “opposing allelic-variances”

Lines 155-156. You run the risk of overpurging by retaining only scaffolds with Arthropoda identifiers in Blobtools. Some unclassified scaffolds might belong to B. brevicornis but are repeat-rich or don’t have strong matches to Arthropoda regions. This is unlikely to affect the conclusions of the paper.

Lines 166, 168. There are two different spellings used for Diachasma (alloeum/alloem). It appears that alloeum is correct and so authors should check for misspelled species names.

Lines 193, 195. The use of g7607 and scaffold 12 is unnecessary. Authors should rewrite here to make it more of a general workflow of going from a query protein to the identification of candidate orthologs. For example, you could change lines 194-195 to “A 20 kbp (or however large it is) region containing a candidate fem ortholog was annotated using HMM plus…”

Lines 195. Authors are inconsistent with spacing of numbers and units (10Kbp, later 123 Mbp is used)

Lines 227. The authors may wish to consider making available all code used in bioinformatics analyses.

Lines 244-245. The listing of the exact genome size seems redundant…you can remove this and say “The final genome is ~123 Mbp, comprised of 353 scaffolds…”

Lines 244-246. Authors should report other standard assembly metrics such as contig N50 that allows readers to get a general characterization of the assembly.

Line 249. Check for consistency issues Mbp vs mbp

Line 262. Rephrase “resemble each other very well”

Line 267. In what sense does the red line in Fig. 1 indicate synteny? Looks like the line indicates a consistency in gene sizes but in no way says anything about co-localization of loci.

Lines 292-293. “level of potential heterozygosity” is awkwardly phrased and there is also no quantitative support for this statement. What is the actual level of heterozygosity in the coding regions of these genes. Why weren’t the phased haplotypes included in the alignments in appendix A?

Lines 292-293. I’m a little concerned about the divergence between the two pseudohaps in Fig 2. Even if you expect heterozygosity to be present in csd/fem regions, wouldn’t overall heterozygosity be low considering the line is highly inbred? You estimated the heterozygosity as 0.54% (line 249) which would be ~100 bp in the ~20 kbp region shown in Fig 2, but in Fig 2 the difference between the sequences looks much higher. The authors should consider additional validation methods to confirm these are true haplotypes and not the result of sequencing/assembly errors. Is there read mapping support for both haplotypes? Is the read mapping depth consistent over the length of these two regions?

Lines 323-332. I think the takeaway message here is that there are a lot of rearrangements present, either due to the long divergence times of the species involved or the inability to recover genes on the same scaffolds due to assembly fragmentation/high protein divergence. The authors should summarize the principle message from this analysis.

Lines 344-346. Is the proximity of csd and fem in A. mellifera sufficient rationale for limiting searches for csd in B. breviconis? Considering a) the fragmentation of assemblies, b) the modest scores resulting from BLAST searches of fem, and c) the evidence of gene rearrangements in this region, it would be prudent to search for csd among all scaffolds? I believe there are many examples of complimentary sex determiners (CSD) in Hymenoptera that do not have exact known locations of the csd gene.

Lines 337-338. Surely the B. brevicornis genome represents a notable contribution to underrepresented braconid genomes, but I wouldn’t consider the gap “filled” as you state here.

Line 343. Since the 10X platform is discontinued, you may wish to highlight alternative methods that researchers could adopt to answer similar questions in the future (e.g. phased assemblies using FALCON).

Line 352. Change to “showed low synteny between”

Line 361. I would change “good csd gene candidate” to “promising csd candidate.” But overall I was a little confused regarding the rationale here. You searched the entire assembly for fem using the A. mellifera protein (lines 191-192) but restricted the search for csd on that same scaffold on the basis that csd is adjacent to fem in A. mellifera (lines 344-346). Why wouldn’t you consider searching all scaffolds here, especially given there are many instances of CSD but most have unresolved molecular causes? Some clarification in the methods would be helpful here.